# Vernier frequency division with dual-microresonator solitons

Beichen Wang 1,4, Zijiao Yang1,2,4, Xiaobao Zhang1,3 & Xu Yi 1,2✉

Microresonator solitons are critical to miniaturize optical frequency combs to chip scale and have the potential to revolutionize spectroscopy, metrology and timing. With the reduction of resonator diameter, high repetition rates up to 1 THz become possible, and they are advantageous to wavelength multiplexing, coherent sampling, and self-referencing. However, the detection of comb repetition rate, the precursor to all comb-based applications, becomes challenging at these repetition rates due to the limited bandwidth of photodiodes and electronics. Here, we report a dual-comb Vernier frequency division method to vastly reduce the required electrical bandwidth. Free-running 216 GHz "Vernier" solitons sample and divide the main soliton's repetition frequency from 197 GHz to 995 MHz through electrical processing of a pair of low frequency dual-comb beat notes. Our demonstration relaxes the instrumentation requirement for microcomb repetition rate detection, and could be applied for optical clocks, optical frequency division, and microwave photonics.

---

[1] Department of Electrical and Computer Engineering, University of Virginia, Charlottesville, VA 22904, USA. [2] Department of Physics, University of Virginia, Charlottesville, VA 22904, USA. [3] College of Advanced Interdisciplinary Studies, National University of Defense Technology, Changsha, Hunan 410073, China. [4] These authors contributed equally: Beichen Wang, Zijiao Yang. ✉email: yi@virginia.edu

Optical frequency combs have revolutionized metrology, time keeping and spectroscopy[1–3], and the past decade has witnessed its miniaturization through optical microresonators[4,5] and dissipated Kerr solitons[6,7]. These solitary wave packets leverage Kerr nonlinearity to compensate cavity loss and to balance chromatic dispersion[8–10]. They output a repetitive pulse stream at a rate set by the resonator roundtrip time, which can range from GHz to THz[11–13]. The reduction of resonator mode volume increases the intracavity Kerr nonlinearity, lowers the operation pump power and extends the comb spectrum span. This has enabled demonstrations of battery-operated soliton combs at 194 GHz repetition rate[14], and octave-spanning soliton generation for self-referencing in a resonator with 1 THz free-spectral-range (FSR)[15]. High repetition rates (rep-rates) are also desired in many comb-based applications. For instance, the maximum acquisition speed in dual-comb spectroscopy[16–18], ranging[19,20], and imaging[21,22], all increase linearly with the comb repetition rate.

However, to detect the high repetition rate, a microresonator-based frequency comb (microcomb) system has to include an auxiliary frequency comb whose repetition rate can be directly detected by a photodiode (PD). The detectable repetition frequency is then multiplied up optically through the equally-spaced comb lines to track the microcombs in action[4,15]. This limits the miniaturization of microcomb system as the area occupied by the resonator scales inverse quadratically with the repetition rate. For the popular electrical K-band, the auxiliary resonator diameter has to exceed several millimeters[23–26]. An approach to divide and detect microcomb repetition frequency beyond photodiode's bandwidth will be critical to eliminate this restriction, and will advance the frequency comb technology in terms of miniaturization, power consumption and ease of integration.

In this article, we introduce a Vernier frequency division method to detect soliton microcomb repetition rate well above the electrical bandwidth in use. In contrast to the conventional approaches, the Vernier frequency division does not require low-rate frequency combs. Instead, the rate of the auxiliary combs, $f_{r2}$, can be higher than that of the main combs, $f_{r1}$, and it can be free-running and stay unknown. The concept is illustrated in Fig. 1. The main and Vernier soliton comb lines create two free-running graduation markings on the optical frequency domain, and similar to a Vernier caliper, these markings coarsely align periodically. Detectable frequency beat notes can be created when the frequency of the $N$-th higher-rate comb line catches up with that of the $(N + 1)$-th lower-rate comb line. These beat notes can be utilized to divide the soliton repetition frequency through an electrical frequency division followed by the subtraction of dual-comb repetition rate difference. Fig. 1 presents one conceptual example, where the main soliton repetition rate divided by $N$ can be obtained from the sum of the first beat frequency $\Delta_1$, and the $N$-th beat frequency $\Delta_N$ divided by $N$. $\Delta_N$ denotes the beat frequency between the $N$-th Vernier comb line and its nearest main soliton comb line.

## Results

The Vernier division reduces the required electrical bandwidth for rep-rate detection from the soliton repetition rate to approximately the repetition rate difference between the main and Vernier solitons, which can be coarsely controlled in microfabrication. In our demonstration, the electrical bandwidth is reduced from 197 GHz to 20s GHz. The Vernier method directly applies to 100s GHz to THz rate soliton microcombs, which are common in many material systems, such as $Si_3N_4$[27–30], silicon[31], AlN[32], and $LiNbO_3$[33–35]. For a fixed electrical bandwidth and rep-rate difference, a higher main soliton rep-rate will

demand a broader comb span in the Vernier method. This is because the number of comb lines required for the comb line frequency of Vernier solitons to overtake that of the main solitons increases linearly with the main soliton repetition rate. At 1 THz repetition rate, 50 comb lines on one side of the pump are needed for 20 GHz rep-rate difference, and this comb span has been reported previously[12,13]. The Vernier division demonstrated in this manuscript could serve as a universal solution for repetition rate detection in various microcomb systems and applications.

In this experiment, the main and Vernier solitons are generated in bus-waveguide coupled $Si_3N_4$ microresonators[36], which have FSRs of 197 GHz and 216 GHz, intrinsic quality factors of $1.5 \times 10^6$ and $2.2 \times 10^6$, and loaded quality factors of $1.3 \times 10^6$ and $1.8 \times 10^6$, respectively. To generate single soliton states, a rapid laser frequency sweeping method[37] is implemented, in which the pump laser is derived from the first phase modulation sideband of a continuous wave (cw) laser, and the sideband frequency can be rapidly tuned by a voltage controlled oscillator (VCO). The pump laser is then split and amplified to generate solitons in both microresonators simultaneously. Thermoelectric coolers (TECs) are used for both the main and Vernier resonators to coarsely align their resonance frequencies at the modes that are being pumped. The complete experimental setup is shown in Fig. 2. Details of the soliton generation is included in the Methods section. Dual-microcomb driven by one pump laser has been previously reported in two cascaded resonators[38], and in a single resonator by counter-propagating and co-propagating pump lasers[39–41].

The optical spectra of single soliton states for main (red) and Vernier (blue) resonators are shown in Fig. 3a. A zoomed-in panel shows the optical spectra where the frequency of the $N$-th Vernier soliton comb line coarsely aligns with that of the $(N + 1)$-th main soliton comb line. An electrical spectrum of the beat frequencies between the two combs is shown in Fig. 3b. Within the 26 GHz cut-off frequency of our electrical spectrum analyzer (ESA), four beat frequencies are observed: $\Delta_1 = 19.3639$ GHz, $\Delta_9 = 22.6815$ GHz, $\Delta_{10} = 3.3157$ GHz and $\Delta_{11} = 16.0449$ GHz. The strong $VCO_1$ beat note near 14 GHz is derived from the modulation of the cw laser, and can be removed by an optical or electrical filter.

Beat frequencies $\Delta_9$ and $\Delta_{11}$ are selected for the main soliton rep-rate division. $\Delta_9(\Delta_{11})$ is the beat frequency between the 9 (11)-th Vernier soliton comb line and the 10 (12)-th main soliton comb line, where $\Delta_9 = 10f_{r1} - 9f_{r2}$, and $\Delta_{11} = 11f_{r2} - 12f_{r1}$. In the measurement, after combining the main and Vernier solitons with a fiber coupler, a bandpass filter is used to pass the comb lines associated with $\Delta_9$, $\Delta_{10}$, and $\Delta_{11}$ for optical amplification. Then a second fiber coupler splits the power into two optical paths, where in each path a bandpass filter is used to select the comb lines of $\Delta_9$ or $\Delta_{11}$, and the corresponding beat note is created on a photodiode. To divide the main soliton rep-rate, $\Delta_9$ and $\Delta_{11}$ are divided by 36 and 44 in frequency, respectively, and sent to a RF mixer to produce their sum frequency, $f_v = \Delta_9/36 + \Delta_{11}/44 = f_{r1}/198$, which is the main soliton repetition rate divided by 198. The electrical spectra of $\Delta_9/36$, $\Delta_{11}/44$ and their sum $f_v$ are shown in Fig. 3c–e. The complete experimental setup is shown in Fig. 2. More experimental details are included in Methods section. In principle, one can use the configuration in Fig. 1, where $\Delta_1$ is mixed with $\Delta_N/N$ to generate $f_{r1}/N$. However, limited by the selection of electrical mixers in our lab, we do not have the capability to mix $\Delta_1$ (~20 GHz) and $\Delta_N/N$ (~2 GHz for $N = 9, 11$), and thus we select $\Delta_9$ and $\Delta_{11}$ instead.

To validate the Vernier method, a conventional method by using electro-optics modulation (EOM) frequency comb is implemented as an out-of-loop verification. In the conventional EOM method, two adjacent comb lines from the main solitons

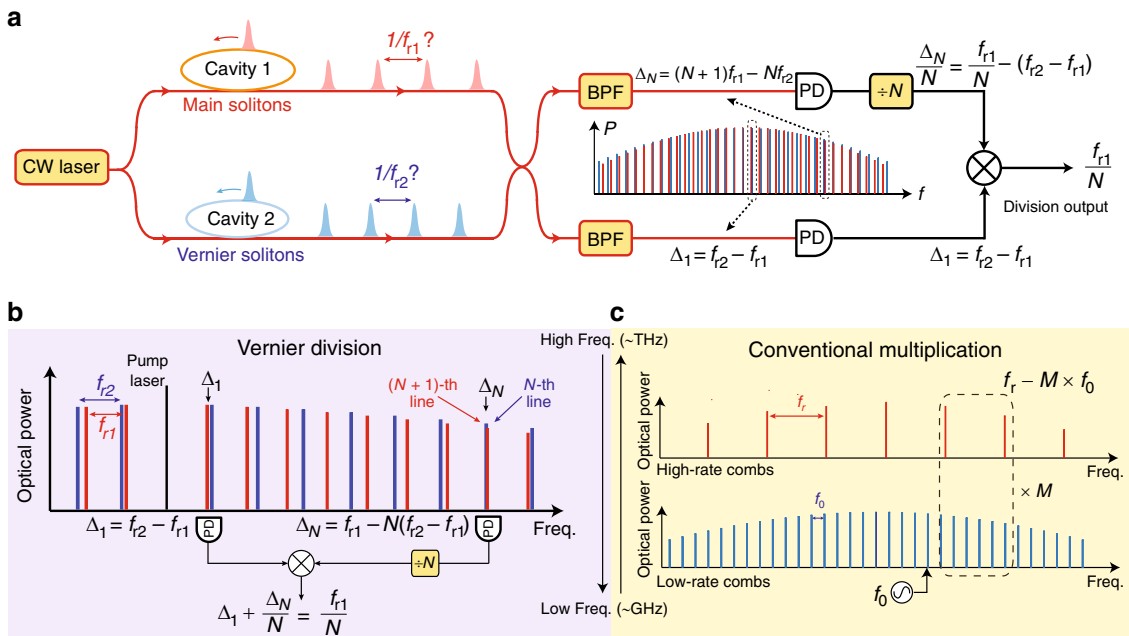

**Fig. 1 Concept of Vernier dual-comb repetition rate division. a** To divide and detect the main soliton (red) repetition rate, a free-running higher rate microcomb (Vernier, blue) is generated to sample and divide down the main soliton rep-rate. Two pairs of low frequency dual-comb beat notes are selected by optical bandpass filters (BPFs) and detected on photodiodes (PDs) to extract the high repetition frequency. **b** The zoomed-in optical spectra to illustrate the Vernier division principle. When the Vernier soliton rep-rate is slightly higher than the main soliton rep-rate, the frequency of the $N$-th Vernier comb line can coarsely align with the $(N + 1)$-th main soliton comb line. The corresponding beat frequency contains information of the absolute repetition rate ($f_{r1}$) and the repetition rate difference ($f_{r2} - f_{r1}$). The main soliton repetition rate can be divided down by $N$ by electrically dividing $\Delta_N$ by $N$, and then adding it with $\Delta_1$. **c** In comparison, conventional repetition rate detection methods require a low rep-rate comb to optically multiply a low frequency reference to a high frequency, which is then compared to the high repetition rate through heterodyne detection.

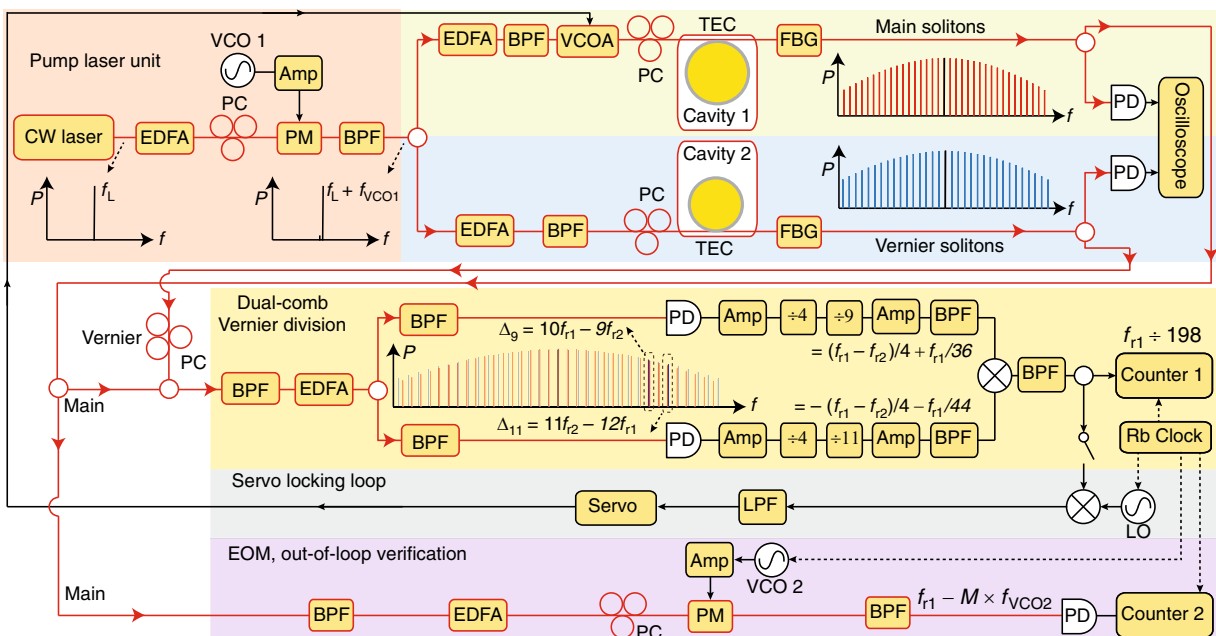

**Fig. 2 Experimental setup.** The main solitons and Vernier solitons are generated in two SiN resonators which are temperature controlled by thermoelectric coolers (TECs). The pump laser is the first modulation sideband of a phase modulated (PM) continuous wave (cw) laser, and the sideband frequency can be rapidly tuned by a voltage controlled oscillator (VCO)[37]. The frequencies of the cw laser and phase modulation are $f_L$ and $f_{VCO1}$, respectively. The main and Vernier solitons are combined and then split to two paths, and two optical bandpass filters (BPFs) are used to select the 9-th and the 11-th pairs of comb lines in each path, respectively. Beat notes $\Delta_9$ and $\Delta_{11}$ are generated by photodiodes (PDs) and they are electronically divided by 36 and 44, respectively. The sum of the two signals is created by a frequency mixer, and its frequency $f_v$ is recorded on a counter. For stabilizing the rep-rate of main solitons, $f_v$ is mixed with a rubidium-referenced local oscillator (LO) to servo control a voltage controlled optical attenuator (VCOA) for repetition rate tuning. For out-of-loop verification, electro-optics modulation (EOM) method is used and shown in the purple panel. Erbium-doped fiber amplifiers (EDFAs), polarization controllers (PCs), electrical amplifiers (Amps), low pass filters (LPFs) and rubidium (Rb) clock are also used in the experiment.

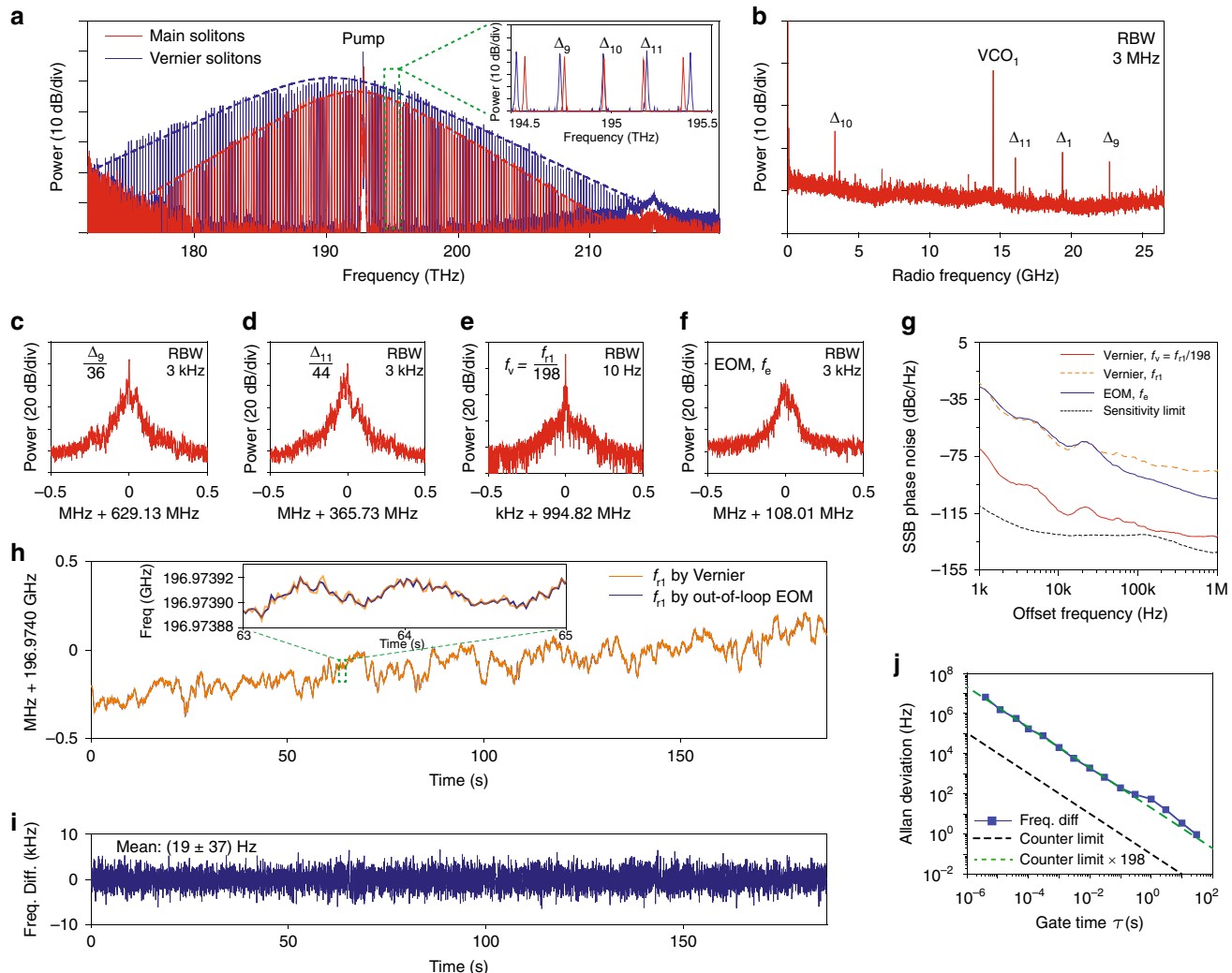

**Fig. 3 Summary of experimental data. a** Optical spectra of main solitons (red) and Vernier solitons (blue) with sech$^2$ envelopes (dashed lines). The 9-th and 11-th pairs of comb lines are shown in the zoomed-in panel. The pump laser is suppressed by Bragg-grating filters. **b** ESA spectra of dual-comb beat notes. $\Delta_1$, $\Delta_9$, $\Delta_{10}$, and $\Delta_{11}$ are apparent. The strong VCO$_1$ beat note is derived from the pump laser unit, and can be filtered out optically or electronically. ESA spectrum of: **c** $\Delta_9$ divided by 36, **d** $\Delta_{11}$ divided by 44, **e** $f_v = f_{r1}/198$ as the sum of $\Delta_9/36$ and $\Delta_{11}/44$, and **f** beat note $f_e$ from out-of-loop EOM method. **g** Phase noise measurement of $f_v$ (red) and $f_e$ (blue). The phase noise of $f_v$ multiplied by 198$^2$ matches that of $f_{r1}$ measured by out-of-loop EOM method. **h** Rep-rate of the main solitons measured by Vernier method (orange) and EOM method (blue). Both main and Vernier solitons are free-running. The gate time is 10 ms. **i** The frequency difference between rep-rate measured with Vernier and EOM methods in panel **h**. Mean value is concluded with a 95% confidence interval under normal distribution. **j** Allan deviation of the frequency difference. The frequency difference agrees with the counter resolution limit for the Vernier method.

are phase modulated at the frequency of a VCO to produce modulation sidebands. The strong modulation results in a pair of sidebands near the midpoint of the two comb lines, and they can be optically filtered and detected[27,42] (see Fig. 2, and Methods section: electro-optics modulation (EOM) comb method). The detected EOM beat note (Fig. 3f) has frequency of $f_e = f_{r1} - M \times f_{VCO2}$, where $M$ is the number of modulation sidebands, and $f_{VCO2}$ is the modulation frequency. $M$ and $f_{VCO2}$ are set to 11 and 17.897 GHz in this experiment, respectively. It is worth noting that the Vernier beat note $f_v$ has much narrower linewidth than the EOM beat note $f_e$, which implies that the rep-rate of the main solitons is coherently divided down from 196.974 GHz to 994.82 MHz.

To show the coherent division in the Vernier dual-comb method, the phase noise of the Vernier beat note, $f_v$, and the out-of-loop EOM beat note, $f_e$, are measured with an ESA through direct detection technique (Fig. 3g). For coherent frequency

division, the phase noise of $f_v$ (red trace) should be 198$^2$ lower than the phase noise of the undivided rep-rate, which is measured through the EOM method (blue trace). This is verified in our measurement, as the phase noise of $f_v$ multiplied by 198$^2$ (orange dash trace) agrees very well with the phase noise of $f_e$ at offset frequency up to 30 kHz. Beyond 30 kHz offset frequency, the phase noise of $f_v$ is comparable to the ESA sensitivity limit (black dash trace). At high offset frequency, our phase noise measurement might be affected by relative intensity noise (RIN). This is common for direct detection technique, as the RIN cannot be separated from the phase noise in the measurement.

The rep-rate of the main solitons can be derived by multiplying the Vernier beat note, $f_v$, by 198. A zero-dead-time frequency counter is used to record $f_v$. The main soliton rep-rate, $f_{r1} = 198 \times f_v$, is shown in Fig. 3h (orange trace). The free-running main solitons have repetition rate around 196.9740 GHz, and the rate is drifting due to temperature and pump laser frequency

fluctuations. This rep-rate measurement is compared to the rep-rate measured with out-of-loop EOM method. The frequency of the EOM beat note $f_e$ is recorded on a second zero-dead-time counter, and the rep-rate is derived as $f_{r1} = f_e + M \times f_{VCO2}$. The EOM-measured rep-rate is shown in Fig. 3h (blue trace), and it overlaps with the rep-rate measured by Vernier method perfectly. The frequency difference between the Vernier-measured rep-rate and EOM-measured rep-rate is calculated and shown in Fig. 3i, and it has a mean value of (19 ± 37) Hz with a 95% confidence interval under normal distribution. Figure. 3j shows the Allan deviation of this frequency difference at various gate times, and it agrees with the counter resolution limit at the frequency of $f_v$ (dash black trace) multiplied by 198 (green dash trace), which is the counter limit for $f_{r1} = 198 \times f_v$. This indicates that no frequency difference between the Vernier method and the EOM method can be detected within the sensitivity of our instruments. In all frequency measurements, the counters and VCOs are synchronized to a rubidium clock.

The main soliton repetition rate can be stabilized by locking the Vernier beat note $f_v$ to a radio-frequency reference. In this demonstration, $f_v$ is locked to a rubidium-stabilized local oscillator through servo control of the pump power using an voltage-controlled optical attenuator (VCOA) to vary the main soliton repetition rate (see Fig. 2). Rep-rate measurement with the EOM method is utilized to verify the locking and the result is shown in Fig. 4a. To eliminate the relative frequency drifts of the electronic components, $f_{VCO1}$, $f_{VCO2}$, counter 1 and counter 2 are all synchronized to the same rubidium clock. Therefore, the error in the rubidium clock has been corrected, and the absolute stability of the reference will not affect our frequency readouts. This allows us to evaluate the servo locking loop without using high performance atomic clock reference. The locking is turned on at the time near 50 s, and the soliton rep-rate immediately stops drifting and is stabilized to 196,962,681,959 Hz (see Fig. 4a). The Allan deviations of the free-running (red) and stabilized (green) rep-rate are calculated from the EOM-based rep-rate measurements and are presented in Fig. 4b. Above 0.3 ms gate time, the Allan deviation of the locked rep-rate scales as $1/\tau$, where $\tau$ is the gate time. Below 0.3 ms gate time, the Allan deviation of the rep-rate follows that of the free-running rep-rate. This behavior of the Allan deviation is expected for a phase-locked oscillator with ~ kHz locking bandwidth. Ultimately, the absolute stability of the rep-rate is limited by the atomic clock reference. It is worth noting that the repetition rate of the Vernier solitons is not stabilized in the entire measurement.

## Discussion

In summary, we have demonstrated the Vernier frequency division method to detect and stabilize soliton repetition rate at 197 GHz with 20s GHz bandwidth photodiodes and electronics. The Vernier method shall be applicable for a wide range of repetition frequencies. It also applies to the case where the two frequency combs do not share the same pump frequency/center frequency. In this situation, one more pair of beat frequency should be detected. As this additional beat note and the two Vernier beat notes share the same offset frequency between the two pump lasers, the offset frequency can be eliminated by frequency subtraction. This will enable the Vernier method to be applied to other types of high-rate combs, such as mode-locked semiconductor lasers[43]. The concept of Vernier dual combs could also be modified to assist carrier-envelope offset frequency ($f_{CEO}$) detection for self-referencing an octave-spanning microcomb. At 1 THz rep-rate, the $f_{CEO}$ given by the f-2f signal can range from 0 to 500 GHz, and it is challenging to keep this frequency in a detectable range as it is subject to small fabrication variations. However, if a Vernier comb is frequency doubled and beat against the main comb, a series of f-2f beat frequencies can be created. Their spacing equals to the dual-comb rep-rate difference, and this can bring the f-2f signal to a detectable frequency. Finally, the Vernier method has the potential to revolutionize optical and electrical frequency conversion by eliminating the need for a detectable repetition rate frequency comb, and it will have direct applications in optical clock[44], optical frequency division[45], and microwave frequency synthesis[26].

## Methods

**Experimental setup of soliton microcombs**. The complete experimental setup is shown in Fig. 2. To overcome the thermal complexity in soliton generation process, the first phase-modulated sideband from a continous wave (CW) laser is used as a rapid-tuning pump laser. The phase modulator is driven by a voltage-controlled oscillator (VCO). The first sideband from the phase modulation is selected by an optical tunable bandpass filter (BPF). With the fast ramp voltage on the VCO, the pump laser scans at a speed of ~20 GHz/μs. A 50/50 splitter after the BPF splits the pump laser equally into two erbium-doped fiber amplifiers (EDFAs). The polarization is carefully adjusted by a polarization controller after each EDFA. The pump laser is coupled into the bus waveguide by a lensed fiber. Single solitons are generated simultaneously in both microresonators by rapidly scanning the pump laser from the blue-detuned regime to the red-detuned regime. The single soliton existence detuning ranges of both microresonators are thermally tuned to overlap. Each microresonator has a temperature controller with 0.01°C resolution. The resonant frequencies are tuned ~2.5 GHz/°C. The main and Vernier solitons are then combined by a fiber coupler. An optical tunable bandpass filter is used to pass three pairs of comb lines, which correspond to $\Delta_9$, $\Delta_{10}$, and $\Delta_{11}$. These comb lines are amplified by an EDFA and then split into two optical paths by a 50/50 fiber coupler. The comb lines corresponding to $\Delta_9$ and $\Delta_{11}$ are then selected by optical

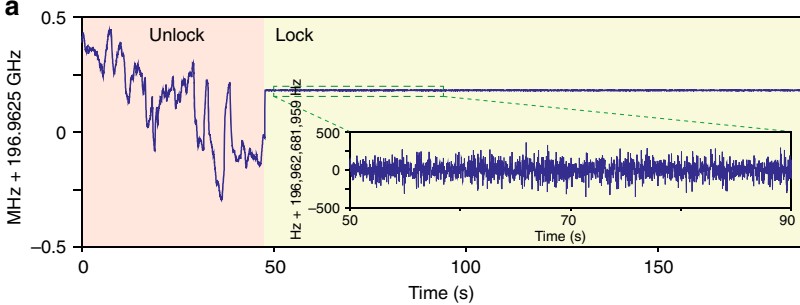

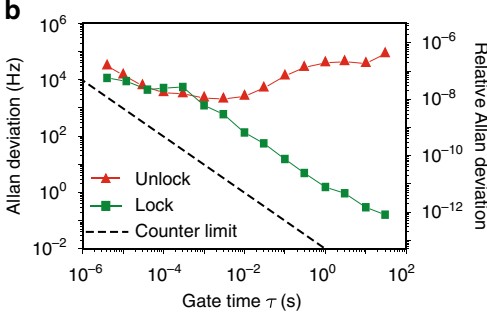

**Fig. 4 Stabilization of main soliton repetition rate by using Vernier dual-comb method.** The rep-rate of the main solitons is stabilized by locking $f_v$ to a Rb-referenced oscillator, and the locking is verified by using EOM method. **a** Rep-rate measurement using EOM method. The locking loop is engaged at the time near 50 s. The gate time ($\tau$) is 10 ms. **b** Allan deviation calculated from the unlocked and locked repetition rates that are measured with the EOM method. The locking loop has ~kHz servo bandwidth. Within the servo bandwidth, the Allan deviation goes down as $1/\tau$. Beyond the servo bandwidth, the Allan deviation is similar to that of the free-running unlock rep-rate. The error in the rubidium clock has been corrected for the Allan deviation of the locked rep-rate. This is done by synchronizing the EOM and the soliton rep-rate to the same rubidium reference. In the entire measurement, the repetition rate of the Vernier solitons is not stabilized, and there is no feedback control of the laser-cavity detuning for the Vernier solitons.

bandpass filters in each path and detected with photodiodes (PDs). The beat notes are amplified to the threshold power of electrical frequency divider for frequency division. Electrical bandpass filters are used to filter out harmonics from dividers, amplifiers, and mixers.

**Vernier frequency division in our experiment**. Vernier frequency division method can use two pairs of comb lines in the overtaking regime, where the frequency of the $N$-th higher-rate comb line catches up with that of the $(N+1)$-th lower-rate comb line. Here, we use the $N$-th pair and the $M$-th pair of comb lines as an example, and $\Delta f_{N,M}$ denotes the frequency difference between the $N(M)$-th Vernier soliton comb line and its nearest main soliton comb line:

$$\Delta f_N = Nf_{r2} - (N+1)f_{r1} = N(f_{r2} - f_{r1}) - f_{r1}, \quad (1)$$

$$\Delta f_M = Mf_{r2} - (M+1)f_{r1} = M(f_{r2} - f_{r1}) - f_{r1}. \quad (2)$$

$f_{r1}$ and $f_{r2}$ are the rep-rates of the main solitons and Vernier solitons, respectively. Eq. (1)/$N$ subtracted by Eq. (2)/$M$ will yield

$$\left(\frac{1}{M} - \frac{1}{N}\right)f_{r1} = \frac{\Delta f_N}{N} - \frac{\Delta f_M}{M}, \quad (3)$$

where the repetition rate of the main solitons, $f_{r1}$, is now expressed by two measurable quantities. In the experiment, photodetecting the corresponding pair of comb lines produces RF signals at the frequency of $\Delta_{M,N}$, where $\Delta_{M,N} = |\Delta f_{M,N}|$. The "$\pm$" ambiguity in $\Delta f_{M,N} = \pm\Delta_{M,N}$ can be resolved by measuring the optical spectra of the main and Vernier solitons.

In our measurement, we select $N=11$ and $M=9$ for the Vernier frequency division. $\Delta_9 = 22.7$ GHz and $\Delta_{11} = 16.1$ GHz are obtained by photodetecting the corresponding pairs of comb lines. These two RF signals are then amplified to ~3 dBm to meet the minimum input power requirement of our frequency dividers. Both $\Delta_9$ and $\Delta_{11}$ are first divided by 4 so that their frequencies are within the frequency bandwidth of the by-9 and by-11 dividers. The output frequencies after division are $\Delta_9/4/9 = 629$ MHz and $\Delta_{11}/4/11 = 366$ MHz, respectively. These two frequencies are then amplified to ~7 dBm and are frequency mixed on an RF mixer. An electrical tunable bandpass filter is used to select the sum of $\Delta_9/36$ and $\Delta_{11}/44$ at the mixer output port. According to Eq. (3), this frequency is equal to $(1/4/9 - 1/4/11)f_{r1} = f_{r1}/198$.

**Electro-optics modulation (EOM) comb method**. In our experiment, part of the main soliton power is sent into the EOM setup for out-of-loop rep-rate verification. The EOM configuration is shown in the purple panel in Fig. 2. An optical bandpass filter is used to select two adjacent comb lines from the main soliton, which are then amplified by an EDFA. They are then sent into an electro-optic phase modulator which is driven by VCO 2 at a frequency of $f_{VCO2}$. Modulation sidebands are created for both comb lines, and when the modulation is strong enough, a pair of sidebands will meet in the midpoint of the two comb lines[42]. This pair of sidebands is then optically filtered by a Bragg-grating filter, and is detected on a photodiode. In our measurement, this EOM beatnote frequency, $f_e$, is ~100 MHz. Using this method, the repetition rate of the main soliton can be derived as $f_{r1} = f_e + M \times f_{VCO2}$, where $M$ is the number of modulation sidebands between the two adjacent comb lines. $M$ and $f_{VCO2}$ are set to 11 and 17.897 GHz in our experiment, respectively. The main soliton repetition rate shown in Fig. 4a is obtained with this method. Allan deviation can then be calculated based on this repetition rate measurement. Ultimately, the correction of the rubidium clock error is limited by the noise added to the EOM sidebands, e.g. residual noise of locking VCO 2 (model: Keysight PSG) to the rubidium reference. These additional noises are not characterized in this experiment.

## Data availability

Source data for Figs. 3 and 4 can be accessed at https://doi.org/10.6084/m9. figshare.12609401. Additional information is available from the corresponding author upon reasonable request.

## Code availability

The codes that support the findings of this study are available from the corresponding authors upon reasonable request.

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

## Acknowledgements

The authors thank Ligentec and VLC Photonics for resonator fabrication, A. Beling and S. Bowers at UVA for the access of signal generator and spectrum analyzer, K. Vahala and Q.F. Yang at Caltech for helpful comments during the preparation of this manuscript, and gratefully acknowledge National Science Foundation (award no. 1842641). X. Y. is also supported by Virginia Space Grant Consortium. X.B.Z is supported by China Scholarship Council.

## Author contributions

X.Y., B.W. and Z.Y. conceived the idea and designed the experiments. B.W. and Z.Y. performed the measurements with the help from X.Z. All authors analyzed the data, participated in preparing the manuscript and contributed to the discussions. X.Y. supervised the project.

## Competing Interests

The authors declare no competing interests.
