## [Peer Review FIle · Nature Communications]

REVIEWER COMMENTS

Reviewer #1 (Remarks to the Author):

In this paper, the authors propose and show a new method for generating an RF signal of the repetition rate of a microcomb divided by a large number. The method involves interfering two pairs of comb lines of one microcomb with a similar comb having a slightly different repetition rate, pumped by the same seed laser. The signal can be used to stabilize the repetition rate of the microcomb. The authors compare the repetition rate measured by their method with a known method of using an EOM to measure the spacing between two comb lines, and find the two methods to be similar in quality. I am not aware of previous work that uses this method.

It is true that microcomb work is often interested in slowing down the repetition rate to maintain compatibility with electronics. In a lab scenario, this method might not be much different than using an EOM, but for practical applications of chip-level systems, this should be quite helpful, as multiple resonators can be built onto a chip.

I can imagine this could be helpful for pushing microcomb work towards THz repetition rates. For a wide audience, this can be meaningful in the sense of making THz spaced combs more accessible, which may find new applications. For the general (low repetition rate) comb community, the technique may not be particularly useful. This type of beat mixing is fairly common for noise reduction or passive stability, and interfering individual comb lines generally requires an intermediate CW laser. There may be some use in dual comb experiments, but otherwise repetition rate measurement is trivial at lower repetition rates.

The writing and available detail is good. Figures are quite detailed, well-labeled, and legible.

One question, there is a line on page 3 using two detectors to improve signal to noise, but it is not clear to me what this is (a balanced detection technique?). Maybe a few more words could be added to the description.

I recommend this manuscript for publication in Nature Communications.

Reviewer #2 (Remarks to the Author):

Microresonator frequency combs (Kerr combs) have seen remarkable progress in the last decade and are now considered for a variety of cutting edge applications, ranging from optical frequency metrology to ultrafast optical ranging.

However, one common challenge is that the repetition rates are so high that they cannot be readout directly using electronics. This challenge needs to be overcome to enable many of the interesting applications.

In this paper the authors show that the readout of a large FSR comb can be done using a Vernier scheme instead of the conventional dual-comb method which relies on a secondary comb with lower, electronically detectable repetition rate. The work is novel and convincing, and the write-up is concise and straight to point.

The authors argue and show that the photodetected beatnote of the first sideband comb line pairs can be electronically mixed with beatnote at the Vernier overlap point divided by an integer factor, resulting in a divided version of the repetition frequency (fr_1/N). This division of fr_1 was analyzed by comparing it with an out-of-loop measurement using an electro-optic modulator (EOM).

Next the authors show that they can stabilize the comb's repetition rate by phase locking the detected beatnote (f_{r1}/N) to a rubidium reference. The authors then use an EOM division / downconversion method to assess the deviations of the comb's repetition rate.

The experiments reported in this paper are of high quality and are likely to have an important impact in advancing Kerr comb applications. I am happy to recommend publication in Nature Communications.

I do have just a few minor comments for authors to address.

In the manuscript the authors mention "see Methods section for EOM configuration," but I couldn't find details there, especially on using the EOM to assess the Allan deviations of the repetition rate. In the conceptual picture they proposed using the first sideband beatnote the overlap beatnote, but in the experiment they've used Δ_9 & Δ_{11} instead. This is probably done to obtain some favorable choice of electrical frequency that fits the BW of their electrical dividers and equipment, however; they should clarify and comment on it. Was the achieved Allan deviation limited by the Rubidium reference or by other sources (e.g. noise from EOM sidebands)? It might be instructive to include the Allan deviation of the Rubidium clock for reference.

Reviewer #3 (Remarks to the Author):

Wang and colleagues present a new method of measurement for frequency comb repetition rates of hundreds of GHz commonly observed in Kerr microresonator frequency comb generation. Using an auxiliary Kerr frequency comb with a comparatively simple degenerate dual pumping setup and a repetition rate frequency mismatch of 10%, they manage to divide the original repetition rate by a factor of 200. They demonstrate coherent division of the 200 GHz repetition rate to below 20 GHz.

Established techniques for high rep rate locking are the use of an auxiliary comb, either a RF rep rate Kerr comb as in Ref. 15 or electro optical comb generation as for example in ref. 37. Both require a low noise independent microwave frequency source and several independent phase locks and are thus at a disadvantage w.r.t. the presented method, which performs rep rate locking with a single phase locking loop. The paper is well written and illustrated with sufficient detail. The extensive literature on dissipative Kerr solitons and is well cited. However, I would like the authors to address the following points, which would improve the readers understanding of the method:

- I suppose the term "free-running" in the caption of Fig.3 means that the laser - cavity detuning is not locked. Please clarify in the caption.
- The frequency division scheme is non-trivial. The authors should explain why the 9th and 11th line were chosen instead of the 1st and 10th line as indicated in Fig. 1. Furthermore, the method section should be expanded and the frequency division scheme (analog/digital) should be explained in more detail.
- "It is worth noting that the Vernier solitons are free-running in the entire measurement, and the rep-rate of Vernier solitons does not need to be stabilized." I guess the laser detuning is meant and/or the repetition rate of the 2nd Kerr comb. Please clarify.
- The authors quote the quality factors of the rings in the text with specifying whether or not the intrinsic or loaded Q factors are referenced and whether the cavities are operated in the under- critically- or overcoupled regime.
- The term "rep-rate", while common in the vocabulary of pulsed laser and frequency comb scientists, should be either formally defined as an abbreviation or avoided.
- The authors should also reconsider the ordering of the figures. The experimental setup figure might be placed 2nd or attached to Figure 1 and the other figures moved down accordingly. I found myself jumping forward in the paper often when reviewing the experimental results in Figs. 2 and 3.

In summary, I support the publication of the presented manuscript in Nature Communications provided that the authors address the raised points.

Reviewer #1 (Remarks to the Author):

In this paper, the authors propose and show a new method for generating an RF signal of the repetition rate of a microcomb divided by a large number. The method involves interfering two pairs of comb lines of one microcomb with a similar comb having a slightly different repetition rate, pumped by the same seed laser. The signal can be used to stabilize the repetition rate of the microcomb. The authors compare the repetition rate measured by their method with a known method of using an EOM to measure the spacing between two comb lines, and find the two methods to be similar in quality. I am not aware of previous work that uses this method.

It is true that microcomb work is often interested in slowing down the repetition rate to maintain compatibility with electronics. In a lab scenario, this method might not be much different than using an EOM, but for practical applications of chip-level systems, this should be quite helpful, as multiple resonators can be built onto a chip.

I can imagine this could be helpful for pushing microcomb work towards THz repetition rates. For a wide audience, this can be meaningful in the sense of making THz spaced combs more accessible, which may find new applications. For the general (low repetition rate) comb community, the technique may not be particularly useful. This type of beat mixing is fairly common for noise reduction or passive stability, and interfering individual comb lines generally requires an intermediate CW laser. There may be some use in dual comb experiments, but otherwise repetition rate measurement is trivial at lower repetition rates.

The writing and available detail is good. Figures are quite detailed, well-labeled, and legible.

One question, there is a line on page 3 using two detectors to improve signal to noise, but it is not clear to me what this is (a balanced detection technique?). Maybe a few more words could be added to the description.

I recommend this manuscript for publication in Nature Communications.

Reply: We thank the reviewer for his/her comments.

The sentence that the reviewer refers to is: “To improve the signal to noise ratio, two pairs of comb lines associated with $\Delta 9$ and $\Delta 11$ are optically filtered and amplified, and detected on two separate photodiodes. ”

We did not use special detection techniques (e.g. balanced detection) to improve the signal to noise ratio. The signal to noise improvement results from optically filtering out the comb lines not associated with $\Delta 9$ and $\Delta 11$, so that the photodiodes will not be saturated by these comb lines. This allows the comb lines associated with $\Delta 9$ and $\Delta 11$ to be amplified to higher power, and thus improve the signal to noise.

This was not well explained in the original manuscript, and we have removed the phrase “improve the signal to noise”, and added the experimental details in the revised manuscript:

“In the measurement, after combining the main and Vernier solitons with a fiber coupler, a bandpass filter is used to pass the comb lines associated with Δ_9 , Δ_{10} , and Δ_{11} for optical amplification. Then a second fiber coupler splits the power into two optical paths, where in each path a bandpass filter is used to select the comb lines of Δ_9 or Δ_{11} , and the corresponding beat note is created on a photodiode.”

Reviewer #2 (Remarks to the Author):

Microresonator frequency combs (Kerr combs) have seen remarkable progress in the last decade and are now considered for a variety of cutting edge applications, ranging from optical frequency metrology to ultrafast optical ranging.

However, one common challenge is that the repetition rates are so high that they cannot be readout directly using electronics. This challenge needs to be overcome to enable many of the interesting applications.

In this paper the authors show that the readout of a large FSR comb can be done using a Vernier scheme instead of the conventional dual-comb method which relies on a secondary comb with lower, electronically detectable repetition rate. The work is novel and convincing, and the write-up is concise and straight to point.

The authors argue and show that the photodetected beatnote of the first sideband comb line pairs can be electronically mixed with beatnote at the Vernier overlap point divided by an integer factor, resulting in a divided version of the repetition frequency (fr_1/N). This division of fr_1 was analyzed by comparing it with an out-of-loop measurement using an electro-optic modulator (EOM).

Next the authors show that they can stabilize the comb’s repetition rate by phase locking the detected beatnote (fr_1/N) to a rubidium reference. The authors then use an EOM division / downconversion method to assess the deviations of the comb’s repetition rate.

The experiments reported in this paper are of high quality and are likely to have an important impact in advancing Kerr comb applications. I am happy to recommend publication in Nature Communications.

I do have just a few minor comments for authors to address.

1. *In the manuscript the authors mention “see Methods section for EOM configuration,” but I couldn’t find details there, especially on using the EOM to assess the Allan deviations of the repetition rate.*

Reply: We have added a section of “Electro-optics modulation (EOM) comb method” in the Methods section, where we have included details of the EOM configuration, and the EOM repetition rate measurement. Details of Allan deviation measurements are added in the main text.

2. *In the conceptual picture they proposed using the first sideband beatnote the overlap beatnote, but in the experiment they’ve used Δ_9 & Δ_{11} instead. This is probably done to obtain some favorable choice of electrical frequency that fits the BW of their electrical dividers and equipment, however; they should clarify and comment on it.*

Reply: The selection of the 9th and 11th lines are a result of the limited frequency range of our electrical mixer. The first sideband beatnote frequency is 19.4 GHz, and the overlap beatnote, e.g. Δ_9 , is at 22.7 GHz, and thus Δ_9 divided by 9 will be around 2.5 GHz. The frequency difference between the first sideband beatnote and $\Delta_9/9$ is relatively large, and we do not have an electrical mixer to mix 19.4 GHz and 2.5 GHz. Therefore, we select $\Delta_9/36$ and $\Delta_{11}/44$, as their frequencies are much closer (629 MHz and 366 MHz), and can be directly mixed in an off-the-shelf electrical mixer. We also added in the Methods section that Δ_9 and Δ_{11} are first divided by 4 so that the frequencies will fall in the bandwidth of our electrical by-9 and by-11 divider.

We have added this discussion in the paragraph where we first introduce $\Delta_9/36$ and $\Delta_{11}/44$:

“In principle, one can use the configuration in Fig. 1, where Δ_1 is frequency mixed with $\Delta N/N$ to generate $f_{r,1}/N$. However, limited by the selection of electrical mixers in our lab, we do not have the capability to mix Δ_1 (~ 20 GHz) and $\Delta N/N$ (~2 GHz for $N=9,11$), and thus we select Δ_9 and Δ_{11} instead.”

3. *Was the achieved Allan deviation limited by the Rubidium reference or by other sources (e.g. noise from EOM sidebands)? It might be instructive to include the Allan deviation of the Rubidium clock for reference.*

Reply: In our measurement, the error from the rubidium clock reference has been corrected. This is done intentionally by synchronizing the EOM and the soliton rep-rate to the same rubidium clock. In effect, the stability of the Rb clock will not limit our Allan deviation result. This allows us to evaluate the rep-rate servo locking loop without using a high performance atomic clock reference. Indeed, as our Rb-clock is an entry level model (Stanford Research Systems, FS725, Allan deviation of 2×10^{-11} and 1×10^{-11} at gate time 1s and 10s), the correction allows us to clearly show that the Allan deviation goes down as one over the gate time ($1/\tau$) within the servo bandwidth.

Our schematic does not correct the noise added in the EOM, such as the residual noise of locking VCO2 to the rubidium reference (locked internally in VCO2, model: Keysight, PSG), RF amplifier noise and etc. However, since the PSG is a much better oscillator than our soliton at free-running, we do not think the noise from the EOM is limiting our Allan deviation measurement. Currently, the Allan deviation of our locked rep-rate is limited by (1) the temperature and mechanical instability of our setup, which causes poor stability beyond the servo bandwidth, and (2) the relatively slow servo locking loop, which cannot correct frequency error above \sim kHz bandwidth.

In the revised manuscript, we have emphasised that the error of the rubidium has been corrected in the figure caption and in the main text. We have added a substantial amount of details to make sure that we don't confuse, or mislead the audience regarding our Allan deviation measurement:

a. We added the Rb clock in the setup figure (figure 2 in the revised manuscript) to clearly show what is referenced to the Rb clock.

b. In the caption of figure 4, we added:

“The error in the rubidium clock has been corrected for the Allan deviation of the locked rep-rate. This is done by synchronizing the EOM and the soliton rep-rate to the same rubidium reference.”

c. In page 5, the paragraph of rep-rate locking, we added:

“To eliminate the relative frequency drifts of the electronic components, f_{VCO1} , f_{VCO2} , counter 1 and counter 2 are all synchronized to the same rubidium clock. Therefore, the error in the rubidium clock has been corrected, and the absolute stability of the reference will not affect our frequency readouts. This allows us to evaluate the servo locking loop without using high performance atomic clock reference. This Allan deviation behavior is expected for a phase-locked oscillator with \sim kHz locking bandwidth. Ultimately, the absolute stability of the rep-rate is limited by the atomic clock reference.”

d. In the Methods section for the EOM method, we added a sentence:

“Ultimately, the correction of the rubidium clock error is limited by the noise added to the EOM sidebands, e.g. residual noise of locking VCO 2 (model: Keysight PSG) to the rubidium reference. These additional noises are not characterized in this experiment.”

Reviewer #3 (Remarks to the Author):

Wang and colleagues present a new method of measurement for frequency comb repetition rates of hundreds of GHz commonly observed in Kerr microresonator frequency comb generation. Using an auxiliary Kerr frequency comb with a comparatively simple degenerate dual pumping setup and a repetition rate frequency mismatch of 10%, they manage to divide the original repetition rate by a factor of 200. They demonstrate coherent division of the 200 GHz repetition rate to below 20 GHz. Established techniques for high rep rate locking are the use of an auxiliary comb, either a RF rep rate Kerr comb as in Ref. 15 or electro optical comb generation as for example in ref. 37. Both require a low noise independent microwave frequency source and several independent phase locks and are thus at a disadvantage w.r.t. the presented method, which performs rep rate locking with a single phase locking loop.

The paper is well written and illustrated with sufficient detail. The extensive literature on dissipative Kerr solitons is well cited. However, I would like the authors to address the following points, which would improve the readers understanding of the method:

1. I suppose the term “free-running” in the caption of Fig.3 means that the laser – cavity detuning is not locked. Please clarify in the caption.

Reply: The laser-cavity detuning is indeed not locked. We have added this sentence in the caption:

“In the entire measurement, the repetition rate of the Vernier solitons is not stabilized, and there is no feedback control of the laser-cavity detuning for the Vernier solitons.”

2. The frequency division scheme is non-trivial. The authors should explain why the 9th and 11th line were chosen instead of the 1st and 10th line as indicated in Fig. 1. Furthermore, the method section should be expanded and the frequency division scheme (analog/digital) should be explained in more detail.

Reply: The selection of the 9th and 11th lines are a result of the limited frequency range of our electrical mixer. The first sideband beatnote frequency is 19.4 GHz, and the overlap beatnote, e.g. $\Delta 9$, is at 22.7 GHz, and $\Delta 9$ divided by 9 will be around 2.5 GHz. This frequency difference, 19.4 GHz vs 2.5 GHz, is very large, and we do not have an electrical mixer to mix them. Therefore, we select $\Delta 9/36$ and $\Delta 11/44$, as their frequency is much closer (629 MHz and 366 MHz), and can be directly mixed on an off-the-shelf electrical mixer.

We have added this discussion in the paragraph where we first introduce $\Delta 9/36$ and $\Delta 11/44$:

“In principle, one can use the configuration in Fig. 1, where $\Delta 1$ is frequency mixed with $\Delta N/N$ to generate $f_{r,1}/N$. However, limited by the selection of electrical mixers in our lab, we do not have the capability to mix $\Delta 1$ (~ 20 GHz) and $\Delta N/N$ (~2 GHz for $N=9,11$), and thus we select $\Delta 9$ and $\Delta 11$ instead.”

In addition, we have added a paragraph in the Methods section to include the details of the frequency division schematic.

3. *“It is worth noting that the Vernier solitons are free-running in the entire measurement, and the rep-rate of Vernier solitons does not need to be stabilized.” I guess the laser detuning is meant and/or the repetition rate of the 2nd Kerr comb. Please clarify.*

Reply: The reviewer is correct. We meant the repetition rate of the 2nd Kerr comb (Vernier soliton) is free running, and is not stabilized to any reference.

We have changed the sentence into:

“It is worth noting that the repetition rate of the Vernier solitons is not stabilized in the entire measurement.”

4. *The authors quote the quality factors of the rings in the text with specifying whether or not the intrinsic or loaded Q factors are referenced and whether the cavities are operated in the under- critically- or overcoupled regime.*

Reply: The intrinsic and loaded quality factors of the main microresonator are 1.5×10^6 and 1.3×10^6 , respectively. The intrinsic and loaded quality factors of the Vernier microresonator are 2.2×10^6 and 1.8×10^6 , respectively. Both resonators are under-coupling. We have included this information in the revised manuscript.

5. *The term “rep-rate”, while common in the vocabulary of pulsed laser and frequency comb scientists, should be either formally defined as an abbreviation or avoided.*

Reply: We have added the definition in the first paragraph, where the repetition rate is first introduced.

6. *The authors should also reconsider the ordering of the figures. The experimental setup figure might be placed 2nd or attached to Figure 1 and the other figures moved down accordingly. I found myself jumping forward in the paper often when reviewing the experimental results in Figs. 2 and 3.*

Reply: We have moved the experimental setup to figure 2 in the revised manuscript.

In summary, I support the publication of the presented manuscript in Nature Communications provided that the authors address the raised points.

REVIEWERS' COMMENTS:

Reviewer #1 (Remarks to the Author):

The author's change to the paper answered my question about detection. I recommend publication.

Reviewer #2 did not share her/his comments with the authors, and just send a positive recommendation to the editors.

Reviewer #3 (Remarks to the Author):

I thank the authors for considering my comments carefully and would like to recommend the manuscript for publications without further changes.